# Recent Development in the Design of Neoglycoliposomes Bearing Arborescent Architectures

**DOI:** 10.3390/molecules26144281

**Published:** 2021-07-15

**Authors:** Leila Mousavifar, Shuay Abdullayev, René Roy

**Affiliations:** Glycosciences and Nanomaterials Laboratory, Department of Chemistry, Université du Québec à Montréal, P.O. Box 8888, Succ. Centre-Ville, Montréal, QC H3C 3P8, Canada; leilyanmousavifar@gmail.com (L.M.); shuay.abdullayev@gmail.com (S.A.)

**Keywords:** glycolipids, neoglycolipids, dendrimers, liposomes, dendrimersomes, carbohydrates

## Abstract

This brief review highlights systematic progress in the design of synthetic glycolipid (neoglycolipids) analogs evolving from the conventional architectures of natural glycosphingolipids and gangliosides. Given that naturally occurring glycolipids are composed of only one hydrophilic sugar head-group and two hydrophobic lipid tails embedded in the lipid bilayers of the cell membranes, they usually require extraneous lipids (phosphatidylcholine, cholesterol) to confer their stability. In order to obviate the necessity for these additional stabilizing ingredients, recent investigations have merged dendrimer chemistry with that of neoglycolipid syntheses. This singular approach has provided novel glycoarchitectures allowing reconsidering the necessity for the traditional one to two hydrophilic/hydrophobic ratio. An emphasis has been provided in the recent design of modular arborescent neoglycolipid syntheses coined glycodendrimersomes.

## 1. Introduction

A large number of classical therapeutic drugs have limited clinical efficacy due to their constrained capacity to reach the targeted tissues or because they are linked to harmful toxic effects at the large doses required to compensate for these weaknesses. Drug delivery through encapsulation into liposomes has been a real breakthrough in improving the therapeutic index of several drugs, particularly in cancer where liposomes were first applied [1]. Liposomes are nanometer-size nanoparticles, often spherical, capable of incorporating either hydrophobic or hydrophilic molecules within the lipid bilayer or the aqueous cavity, respectively. Several hundreds of drugs, including anticancer and antimicrobial agents, chelating agents, peptide hormones, enzymes, proteins, vaccines, and genetic materials, have been encapsulated into the aqueous or lipid phases of liposomes. One of the most definitive demonstrations of these excellent properties has recently been widely observed with the SARS-CoV-2 vaccines of Pfizer-BioNTech and Moderna encapsulating mRNAs. Liposome technologies have greatly matured and they are now formulated into various sizes, compositions, and other characteristics, including surface groups anchoring capable of specific tissue targeting [1,2,3], such as carbohydrates [4]. In this way, they can selectively deliver to the target site for in vivo applications, thereby, dramatically increasing the therapeutic index of otherwise deleterious drugs

However, aqueous solutions of liposomes face physical and chemical instabilities during long-term storage. Hydrolysis and oxidation of phospholipids and liposome aggregations are the most common cause of their instabilities. Interestingly, carbohydrates have been investigated for their ability to protect liposomes against fusion and leakage during lyophilization processes. A particular aspect has been the discovery that sugars offer cryo-protection [1]. Recent liposomal formulations, such as PEGylated liposomes (stealth-liposomes), can extend blood circulation time and vary drug distribution in the body, which can also reduce possible cardiotoxicity. Several lipid structures are commonly found in modern liposome formulations (Figure 1).

## 2. Natural Glycolipids

Glycolipids are important members of the glycoconjugate family [4]. They are endowed with natural amphiphilicity since they are composed of hydrophilic carbohydrate head groups and lipophilic tails. They are essential molecules amongst biomolecules as they are implicated in many complex biological processes. Several simple representatives are also used as surfactants in detergency or emulsification technology. In the complexity of biological interactions and cell-cell communications, their amphiphilicity is mostly responsible for their physicochemical activities and peculiar functions [5]. For glycolipids of biological relevance, this is associated with their location in, out or within cell membranes and their aptitude to cross it. More importantly, they can associate together on the cell surfaces to form colonies forming rafts that are responsible of fundamental multivalent carbohydrate-protein and carbohydrate-carbohydrate interactions [6,7,8,9,10,11,12,13,14,15] having physiological significances [16]. 

Glycolipid-forming liposomes closely resemble that of cell membranes, albeit greatly simplified. They have been comprehensively studied as models of cell membranes [4]. Reconstitution of the membrane-bound carbohydrates within liposome bilayers has been one of the most useful techniques in studying the functions of the membrane glycoconjugates. Glycoliposomes are attractive as delivery systems due to their capacity to improve the stability, therapeutic efficiency, and pharmacokinetic properties of drugs while reducing their side effects and have the advantage of being biodegradable and nontoxic. In addition, surface chemistry and lipid composition can be easily modified to address their precise applications. Cell surface carbohydrates have specific interactions with their cognate proteins, which play an important role in various biological recognition processes, such as fertilization, metastasis, inflammations, and host–pathogen adhesion. Therefore, they serve as attractive molecules for surface modification of liposomes with purpose for specific biomedical applications.

Glycolipids can be classified based on their lipid moieties as glycoglycerolipids, glycophosphatidylinositols, and glycosphingolipids. Glycolipids are found in membranes of most living organisms and their carbohydrate components are directly involved with their recognition/protection properties [17,18]. Microbial-derived glycolipids are increasingly serving as models for sustainable and stable sources of highly diversified, yet simple and economically viable glycolipids [19]. Together with other glycoconjugate members of cell surface glycoproteins, they form the glycocalyx, which coats the cells that provide contact with their environment. Carbohydrates and their cognate glycolipids and glycoproteins at the cell surfaces are involved in key biological phenomena, such as tumor cell expression or markers, bacterial and viral infections, inflammatory responses, and immune cell regulations. The precise role of the lipid tails of glycolipids, embedded within the lipid bilayers of the cells, is not yet fully understood. Nonetheless, they play a major role in correctly presenting carbohydrate antigens to other cells or to protein receptors, as well as in membrane stability and overall organization. Although the lipid components of eukaryotic glycolipids are mostly built around sphingolipids, those of prokaryotic organisms are much more diversified and complex. 

Analysis of the self-assembling properties of glycolipid analogues has highlighted the crucial role of the conformations and molecular packing of the hydrophobic chains in the understanding of the interfacial aggregation phenomena. The physicochemical properties of artificial lipid architectures, mimicking the natural membrane composition, are informative for their exploitation in designing more stable and viable version of biologically relevant vesicles [20] and for drug delivery [21]. Therefore, optimizing the molecular architectures of glycolipids carbohydrate has been the subject of intense activities [22]. The high polarity and biological recognition patterns of the carbohydrate moieties coupled with the hydrophobic character of the lipid tails are both intimately related and make these two molecular entities responsible for the nanomaterial properties they are endowed with.

## 3. Natural Glycosphingolipids (GSLs) and Gangliosides

More complex glycolipids form a special family identified by the sphingolipids (SLs) and glycosphingolipids (GSLs) members that are common structural components of mammalian cell membrane. Sphingolipids are composed of a characteristic ceramide moiety composed of an *N*-acylated sphingosine group (2-amino-4-*trans*-octadecene-1,3-diol). Glucose or galactose glycosidically linked to the primary hydroxy group of the sphingosine moiety provides the simplest glycosphingolipid family members: Glucosylceramide and galactosylceramide (Figure 2). Linkage of a phosphorylcholine moiety results in sphingomyelin, a very abundant membrane lipid. Further additions of oligosaccharides and sulfate groups to glycosphingolipids give rise to a broad range of complex glycosphingolipids, such as sulfatide and α-galactosylceramide. Those with a capping *N*-acetylneuraminic acid are known as gangliosides. Giving their roles in neurodegenerative diseases, their studies and syntheses have been largely documented [17,18]. 

## 4. Neoglycolipids (NGLs) and Glycan Microarrays

Several strategies for surface glyco-functionalization of liposomes have been reported in the latest decades. In particular, two strategies have been commonly used [4]. The first one is a direct liposome formulation approach. First, it consists of the design of the glycolipid ligand, followed by the preparation of liposomes in combination with other principal lipids, a major weakness that the present review is hoping to emphasize. The second approach, known as the post-functionalization approach, mainly involves grafting the carbohydrate ligand onto the preformed liposomes via various simple and direct ligation chemistries. Biomedical applications of glyco-functionalized liposomes generally, include targeted drugs, genes, and antigens delivery. 

Given the complexity of natural glycolipids, glycosphingosines, and gangliosides in particular, several efforts have been directed at the syntheses of simpler analogs referred to as neoglycolipids [22]. The problems associated with naturally occurring glycolipid syntheses rely on: (i) Difficulty in synthesizing the stereo- and region-controlled side chains; (ii) selective reduction of the sugar protecting groups in the presence of the sphingosine unsaturation, isomerization, racemization and/or hydrolysis. Several groups have already reported synthetic glycolipids [23,24]. 

The biological importance of multivalent and rather specific carbohydrate–protein interactions has triggered a plethora of sophisticated methods for unravelling the exact glyco-architectures of the glyco-ligands involved in the biological processes of interest. However, there are major challenges in translating the identity and characteristics of biological systems that operate through carbohydrate recognition. This is due to the fact that glycans cannot be cloned, complex glycan structures are difficult to synthesize, and oligosaccharides isolated from natural sources in homogeneous form are generally available only in limited quantities. In addition, it is well-recognized that carbohydrate-protein interactions are generally weak. The pioneering work of the Feizi’s group in the creation of neoglycolipids (NGLs) has paved the way for the modern techniques in glycan microarrays [25]. The technology involves conjugation of glycans, pure or as mixtures, to an amino lipid of the phosphatidylethanolamine type to generate artificial glycolipids (Scheme 1). The hydrophobic lipid tails of NGLs with amphipathic properties can be immobilized on solid surfaces for detection of their cognate binding proteins. Liposomal formulation of NGLs can also mimic the glycans displayed at the cell surface with their intrinsic mobility, thus can form high-avidity rafts. Most importantly, NGLs can be incorporated into liposomes in a multivalent state for inhibition assays.

Notably, the amphipathic properties of NGLs account for the absence of non-specific binding on the glycan microarrays or in enzyme-linked lectin/immunoassays (ELLA/ELISA). An additional advantage of liposomal formulation of NGLs is that they can be prepared in aqueous buffers rather than in volatile solvent. Analogously to natural glycolipids, NGLs derived from extracted glycan mixtures can be resolved on high-performance TLC (HPTLC) plates for binding studies. Moreover, the good ionization properties of NGLs facilitate sequences determination of the glycan moieties during mass spectrometry analyses. 

Several other techniques exist toward the syntheses of more simple and unnatural glycolipids. Amongst these, the efforts of the Tiamaki’s group toward the preparation of lipid tails harbored on aromatic scaffolds are notable (Scheme 2) [22]. They were synthesized using typical glycosyl donors, such as the peracetylated mannopyranosyl trichloroacetimidate, and (**1**) aromatic lipid tails (**2**–**4**) using Lewis acid catalyzed glycosidation. The glycolipids could be easily incorporated into liposomes of L-α-phosphatidylcholine which were fully characterized and purified by gel-permeation chromatography. The liposomal formulation could be prepared by both conventional techniques, i.e., the injection method or the thin film method. Uniform and unilamellar vesicles of ~130 nm were readily obtained by several passages on a membrane extruder (100 nm). The sugar moieties of the synthetic glycolipids possessing a hexamethylene spacer (**6**) were clearly accessible on the surface of the liposomes and interacted specifically with their cognate lectins (Ricin for galactoside and Concanavalin A for mannosides) to give liposomal assemblies. The agglutination of the corresponding glycoliposomes induced by lectins was determined by turbidity analyses and particle size based on dynamic light scattering and laser diffraction methods. Liposomes possessing a shorter ethylene (**5**) or longer decamethylene (**7**) linker gave poor lectin-induced agglutinates, indicating that the length of the aglycon linkers were critical in the carbohydrate-lectin interactions. All together, these data formed the basis for the author’s investigation on arborescent glycoliposomes (dendrimersomes) (see below). Furthermore, the stability of the liposomes was determined by fluorometry. After standing for 1 month in HEPES buffer at room temperature, the fluorophore used (calcein) in the internal aqueous phase did not leak, indicating that no collapse of the glycolipid-incorporated liposomes occurred under the conditions used. From dynamic light scattering (DLS) analysis, neither aggregation nor change in the size distribution of the dispersed liposomes was observed. Analogous mannosylated liposomal preparations, incorporating the anti-cancer chemotherapeutic cytarabine (Ara-C) were recently shown to possess immunomodulatory properties [21]. 

An even simpler version for the preparation of a wide range of neoglycolipids was published in the author’s laboratory [26]. It was based on the very efficient and versatile radical induced (AIBN) thiol-yne reaction involving propynyl glycosides, such as compound **8** and a series of alkanethiols (**9, 10**) (Scheme 3). When the reactions were performed in a one-pot process using a slight excess of the alkane thiols, symmetrical thioglyceroglycolipids were obtained (**13**, **16**). More importantly, when the reactions were sequentially accomplished with one equivalent of alkane thiols (**9, 10**) and the intermediate vinylic thioethers isolated (**11, 12**), the second thiol additions could be executed with different alkanethiols to afford unsymmetrical thioglyceroglycolipids (**14, 15**). 

Liposomes of 150–300 nm were obtained by solvent injection of their ethanol or tetrahydrofuran (THF) solution in water. The resulting structures were analyzed by DLS and atomic force microscopy (AFM). The glycosylated lipid nanoparticles showed good stability in water. Alternatively, giant soft unilamellar vesicles were also obtained by film hydration and visualized by differential interference contrast microscopy (DIC). Incorporation of a hydrophobic dye (Nyle Red) to the solution prior to evaporation allowed visualization by confocal microscopy [26]. Finally, the biological functions of the newly formed glycolipid vesicles were evaluated by multivalent carbohydrate–protein binding interactions using concanavalin A (ConA) for the mannolipids. Agglutination assays and the binding of the mannoglycolipid by dendritic cells (DCs) induced an increase in DCs immunostimulatory potential. Importantly, no changes in cells viability at tested doses were observed. Moreover, the mannosylated liposomes were investigated as a potential method to improve the plasma stability of peptide-based drugs, such as the kappa opioid receptor selective antagonist dynantin, and the NOD2 innate immune receptor ligand muramyl dipeptide (MDP). The combination of MDP with the glycolipid led to better peptide entrapment, which greatly improved its plasma stability [27,28].

## 5. Neoglycoliposomes Bearing Arborescent Architectures

The structures of natural glycolipids and neoglycolipids described thus far were essentially built from classical approaches utilizing one hydrophilic sugar head group and two lipid tails, as seen in Figure 1 for liposome precursors. In the next sections, these “standard” molecular architectures will be referred to as being of a 1:2 ratio (one sugar: two lipid tails). In addition, most liposomal formulations based on the above glycolipids also contained these compounds in order to stabilize the vesicles, including cholesterol. Until recently, there have been only scarce examples challenging this avenue [29]. In light of the various successes encountered with glycodendrimers as multivalent display for improved carbohydrate-protein interactions [6,7,8,9,10,11,12,13,14,15], the concept was extended to a new family of glycolipids exposing arborescent architectures. This novel family of neoglycolipids has been coined as “glycodendrimersomes” [30,31,32]. The roles of the newly created arborescent architectures have been to increase the liposomal stability, their binding affinity, the carbohydrate distribution/localization, and to avoid the use of common lipid additives, such as cholesterol and egg yolk components.

One early example was designed to compare the relative “multivalent” efficacy between glycodendron peptides built on the complex sialyl Lewis X (sLe^x^) tetrasaccharide and its liposomal equivalent [33]. First, the sLe^X^ glycodendrons were synthesized on L-lysine scaffolds according to Scheme 4. Initially, the three amine groups of the lysine dimer (**17**) were coupled with the homobifunctional spacer ethylene glycol bis(succinimidylsuccinate) (EGS, **18**) to provide trimeric active ester **19**. After HPLC purification of the EGS-modified peptides with one *N*-hydroxysuccinate function per EGS left unreacted, the trimeric **19** was next grafted to glycylamine derivatives of sLex (**20**) in a second step to give trimeric sLe^X^ (**21**) (dimer not shown). The analogous sLe^x^ glycolipid **25** was prepared using dimyristoylphosphatidylethanolamine (DMPE, **22**) coupled to disuccinimidyl suberate (DSS, **23**) as a homobifunctional spacer to afford intermediate active ester **24**, which was then treated with sLe^x^ glycylamine **20** to give the final sLe^x^ glycolipid **25**. The liposomes were prepared using egg yolk phosphatidylcholine. The vesicle diameters ranged from 36 nm to 97 nm with decreasing sLe^x^-DMPE content.

The set of multivalent sLe^x^ was evaluated using three types of binding assays to block receptor mediated hepatocyte HepG2-cells binding: soluble anti-sLe^x^ monoclonal antibody (CSLEX1); immobilized E-selectin; activated human umbilical vein endothelial cells (HUVECs). Compared to the monovalent sLe^x^ (**20**), the inhibition powers of both sLe^x^ dimer (not shown) and trimer **21** were enhanced up to 50-fold for cell binding to the soluble antibody, and that of sLe^x^-liposomes made from **25** by 7 orders of magnitude, i.e., ~3 × 10^−^^11^ M for 50% inhibition (IC_50_). The inhibition activity against immobilized E-selectin was enhanced only 3-fold for the dimer and 10-fold for the trimer (**21**) but 5 orders of magnitude for sLe^x^-liposomes containing **25**, respectively. A similar tendency was observed in the HUVECs assay. Compared to monovalent sLe^x^ (**20**) used as a reference point, the relative efficiencies of the dimer and the trimer (**21**) were one and two, respectively, but about 20,000 for sLe^x^-liposomes resulting from **25**. It was concluded that the multivalency of the sLe^x^-ligands prepared is an essential but not sufficient precondition for a high inhibition potency. Additionally, the structural properties of inhibitors determine their binding behavior, which must be considered for the design of potential therapeutic probes.

True examples of glycodendrimersomes bearing 2:2 and 4:2 sugars to lipid ratios were described as early as 2008 by the team of Schuber et al. [34]. In that study, the authors evaluated the influence of the sugar valency for the targeting of human dendritic cells using mannosylated liposomes. The development of new generations of vaccines necessitates an efficacious delivery of the antigenic portions to antigen-presenting cells (APCs) such as dendritic cells, known to harbor mannopyranoside receptors (DC-SIGN) [35]. The lipid portion of these compounds was prepared in six steps starting from glycerol, tetraethylene glycol, and oleyl alcohol to afford lipid **26** (Scheme 5). Carbodiimide (DCC, NHS, CH_2_Cl_2_) coupling with peracetylated mannoside **27** gave conventional glycolipid **28**, having a sugar to lipid ration of 1 to 2, after sugar deprotection (K_2_CO_3_, MeOH). Similar treatment with mannosyl dimer (**29**) and tetramer (**30**) afforded neoglycolipid **31** and **32** having the 2:2 and 4:2 sugar to lipid ratios, respectively.

The neoglycoliposomes were prepared by mixing egg yolk phospholipids L-α-phosphatidylcholine (PC), L-α-phosphatidyl-DL-glycerol (PG) and cholesterol (75/20/50 molar ratio), in chloroform with the appropriate mannosylated lipid (**28**, **31**, **32**) at 0–16 mol% mannose content. In their study, the authors compared the interaction of plain (70 nm) and mannose-targeted liposomes (110 nm), containing mono-, di-, and tetraantennary mannosyl lipid derivatives, with human monocyte derived immature dendritic cells (iDCs). 

Efficient mannose receptor-mediated endocytosis by iDCs was observed for the mannosylated liposomes. In contrast, only non-specific interaction with little uptake was observed with plain liposomes. As anticipated, liposomes prepared with multibranched mannosylated lipids (**31**, **32**) displayed higher binding affinity for the mannose receptor than vesicles containing the mono-mannosylated analog (**28**). Interestingly, it was found that di-mannosylated ligands (**31**) present at the surface of the liposomes were as efficient as tetra-mannosylated ones (**32**) in uptake/endocytosis. Importantly, once presented as multivalent glycodendrimersome, high generation dendrimers are not an absolute requirement for the design of vaccines. In addition, the mannose-mediated uptake of liposomes did not result in an activation of iDCs. 

Another investigation by the group of Guo et al. described the synthesis of glycodendrimersomes presenting a ratio of 3:2 of the E-/P- selectin ligands 3′-sulfated Lewis^a^, an important interaction involved in inflammatory processes [36]. For this purpose, they used a pentaerythritol scaffold (34) and the arborescent glycolipid was constructed according to Scheme 6 using compounds **33–38** as precursors. The glycodendrimersomes were prepared using 1,2-distearoyl-*sn*-glycero-3-phosphocholine (DSPC, 55 mol%), cholesterol (40 mol%) and **39** (5 mol%). The sizes of the liposome, with, and without, sugar, after extrusion were 120 and 110 nm, respectively as measured by DLS. The glycoliposomes were stable for at least a month when incorporated with 5 to 15 mol% of the glycolipid.

Self-assembling of arborescent neoglycodendrimersomes harboring α-*C*-linked galactopyranosides mounted on a 2,2-bis(hydroxymethyl)-propionic acid (bis-HMPA) scaffold ending with palmitic acid dimers has been described by the Gillies group [37] (Scheme 7). The glycolipid hybrid was made to mimic the natural KRN 7000 immunostimulant (Figure 2). The synthetic strategy involved the initial preparation of a distearyl glycerol lipid tail (**40**) functionalized with an azide group for further coupling using the CuAAC click chemistry. The glycodendrons, such as the octameric **41** were composed of a series of polyester dendrons of G0–G4 generations having peripheral amine groups and an alkyne functionality at the focal point. The amines were then conjugated to a *C*-linked α-D-galactoside derivative (α-Gal) ending with an isothiocyanate functionality to afford amphiphiles **42** having 2–16 α-Gal moieties, respectively. The structure represents an interesting NGL with an 8:2 ratio of the hydrophilic to hydrophobic tail, respectively. Aqueous self-assembly using the injection method resulted in vesicles for the G0 through G2 generations and micellar structures for the higher G3 and G4 generation. Again, the authors concluded that the hydrophilic–hydrophobic balances were determinant factors for these amphiphilic structures.

A versatile and overwhelming approach to glycodendrimersomes has been described by the Lindhorst’s group [38]. The powerful strategy is based on the post-synthetic transformation of pre-formed glycodendrons having suitable functionalization at the focal point. Given the wide range of available glycodendrons in the literature [6,7,8,9,10,11,12,13,14,15], the methodology would be readily amenable to scalable, complex, and useful liposomes having arborescent glycoarchitectures. An example is depicted in Scheme 8 [38]. It involved the use of readily available carbohydrate derivatives having an alcohol function in the aglyconic moiety, such as in mannoside **43**. Initial treatment with methallyldichloride (3-chloro-2-chloromethyl-1-propene) (**44**) (NaH, THF) gave rise to dimer **45**. Reductive ozonolysis (O_3_, NaBH_4_, CH_2_Cl_2_, MeOH) afforded alcohol **46**, which upon a second coupling with methallyldichloride, provided tetramer **47** in 63% yield. Reductive ozonolysis as above followed by allylation gave **48** (69%), suitable for a cross-metathesis reaction using the ruthenium based Grubbs’ catalyst. Indeed, cross-metathesis with allylated glycerolipid **49** (38%), hydrogenolysis of the resulting *trans*-double bond (H_2_, Pd-C, MeOH, 94%) and full acid-catalyzed acetal deprotection (TFA, H_2_O, 9:1) gave mannosylated glycodendrimersome **50** in 98% yield. 

## 6. Modular Approach to Arborescent Glycoarchitectures

Another widely applicable and adaptable synthetic strategy toward arborescent glycolipids was achieved through the collaboration of the Roy’s and Percec’s groups [30,31,32]. The approach is depicted in Figure 3. It involved a modular combination of subunit fragments based on the judicious choice of the core scaffolds, such as pentaerythritol (**34**) or its equivalent amine branch tris(hydroxymethyl)aminomethane (**TRIS**). This was followed by the build-up of a wide range of lipid moieties elaborated on 3,5-dihydroxybenzoic acid, its 3,4-dihydroxy counterpart, or 3,4,5-trihydroxybenzoic acid (gallic acid). Short PEGylation on suitably branched scaffold next afforded the often necessary hydrophilic branches useful to interspace the sugar residues that are best post-assembled using click chemistry. In their initial settings, the authors constructed 7 libraries composed of 51 self-assembling amphiphilic Janus dendrimers. Their self-assembly by simple injection of THF or ethanol solution into water or buffer and by hydration were ascertained by a combination of methods including dynamic light scattering, confocal microscopy, cryogenic transmission electron microscopy, Fourier transform analysis, and micropipette-aspiration experiments to assess mechanical properties. These assemblies were stable over time in water and various buffers, exhibited narrow molecular-weight distribution, and displayed dimensions that were programmable by the concentration of the solution from which they were injected. These results demonstrated, for the first time, the candidacy of glycodendrimersomes as new mimics of biological membranes with programmable glycan ligand presentations, as supramolecular lectin ligands, and the possibility to use them as vaccines and targeted delivery devices. 

A detailed depiction of the modular synthetic strategies is illustrated in Scheme 9. Three sets of sugar derivatives were used as either propargylated aglycons or as extended azides: Galactosides (Gal) **51, 52**; mannosides (Man) **53, 54**; and lactosides (Lac) **55, 56**. They could be coupled to either of their functional lipid counterparts **57**–**62** using copper-catalyzed azide-alkyne cycloaddition (CuAAC). As stated above, 7 libraries of 51 compounds could be produced using a family of lipids. These libraries revealed a diversity of hard and soft assemblies, including unilamellar spherical, polygonal, and tubular vesicles, aggregates of Janus glycodendrimers, and rodlike micelles, and glycodendrimermicelles, cubosomes, and solid lamellae.

An additional advantage of the synthetic strategy was the capacity to construct various families of architectures that were coined: Single-single; twin-twin; and twin-mixed (Figure 4) [31]. The single-single motif is essentially reminiscent to the one described above in Scheme 2 [21,22], and recently used in drug delivery [21]. The twin-mixed glycodendrimersome topology was shown to be most efficient in binding three different lectin species from plants, bacteria, and humans. This behavior was likely due to better accessibility of the sugar moiety by the large multivalent lectins. In their third paper [32], the authors demonstrated an intriguing property originating from the lactosylated twin-mixed glycodendrimersome against the wild-type human galectin-1 (WT hGal-1). The WT hGal-1 is a homodimeric lectin, the two subunits of which being held together by hydrophobic protein-protein interactions. An engineered version of the hGal-1 was produced in which the two protein subunits were covalently linked together by peptide bonds. When the lactosylated liposomes were allowed to agglutinate the lectins, through the usual cross-linking abilities, the wild-type dimers dissociated into monomeric protein that could no longer be agglutinated over time. These results clearly demonstrated the strong affinity/avidity of such glycodendrimersomes. Examples of the construction of typical twin-mixed structures are illustrated in Scheme 10.

Detailed structures of a family of three different sugars Gal (**51**), Man (**53**), and Lactose (**55, 63**) having aglyconic azide group that was coupled to inter-spaced Pegylated alkyne precursor (**64**) using the click reaction CuAAC are depicted in Scheme 10 to provide twin-mixed glycolipids **65**–**68**. Briefly, propargyl alcohol **69** was treated with succinic anhydride (**70**) to give monoester **71** in 86% yield, which upon treatment with carbodiimide (DCC), gave anhydride **72** (92%). Treatment of TRIS (**73**) with acetone under acidic conditions afforded **74** in 60% yield. Condensation between **72** and **74** provided intermediate **75** in 92% yield. Further carbodiimide coupling with PEGylated gallic acid **76** gave **77** (94%). Acetal methanolysis of **77** under acidic conditions gave diol **78**, which after another round of carbodiimide coupling with lipid acid tail **79** provided key alkyne-bearing intermediate **64** (91%). The syntheses were also extended to the 3′-*O*-sulfated lactoside analog **63 [39]**, a useful modification that allows further selectivity with the family of galectin receptors [40,41,42].

Arborescent neoglycoliposomes synthesized until recently were prepared via copper-catalyzed azide–alkyne cycloadditions (CuAAC) to facilitate the anchoring of the unprotected sugar moieties to the ester-sensitive lipid tails. Janus glycodendrimersomes had hydrophilic linkers contained three to four oligo(oxyethylene)s that were directly connected to the sugar. To expand the scope of these glycoarchitectures, automated solid-phase glycan assembly (AGA) allowed access to more complex glycans. Therefore, oligosaccharides obtained by AGA and equipped with an *N*-pentyl amino hydrophobic linker that facilitates the conjugation to surfaces or biomolecules are readily available. However, these hydrophobic *N*-pentyl amino linker attached to oligosaccharides did not fulfill the most fundamental structural requirement for the construction of arborescent NGLs by the click chemistry already elaborated. Fortunately, ample literature exists demonstrating that the sugar built with the isothiocyanate group in the aglyconic portion could also be directly used for conjugation to amine-containing nanomaterials and proteins via a thiourea functionality. The strategy has been extended to the usual saccharide described above together with oligomannosides bearing either α-(1-2) and α-(1-6) linkages (**69–76**) (Scheme 11) [43]. The resulting Janus glycodendrimersomes, prepared in this way, enabled the binding of the glycans to their associated lectins, as previously. 

## 7. Arborescent Neoglycolipids as Detergents

The study of integral membrane-bound protein structures and functions is troublesome due to the difficulty in extracting and handling these highly lipophilic proteins. Classical examples are the family of G protein coupled receptors (GPCRs). Aqueous solubilization, necessary for common biophysical analysis (ex. NMR), generally requires a detergent to shield the large lipophilic surfaces of the native proteins. Detergents are valuable tools for membrane protein manipulation. The micellar aggregates formed by detergent have the ability to encapsulate the hydrophobic domains of membrane proteins. The resulting protein−detergent complexes become compatible with the aqueous environments, making structural and functional analyses feasible.

Many proteins remained difficult to investigate due to the lack of suitable detergents. Chae and coworkers have introduced a class of arborescent amphiphiles with hydrophilic groups derived from maltose (**77**, **78**) (Scheme 12) [44]. Representatives of this maltose–neopentyl glycol (MNG) amphiphile family showed favorable behavior relative to conventional detergents, as manifested in multiple membrane protein systems, leading to enhanced structural stability and successful crystallization. Therefore, MNG neoglycoamphiphiles represent promising tools for the structural analyses of membrane-bound protein. Moreover, they recently also disclosed improve families of these interesting detergents as exemplified by the mannitol-based amphiphilic compound **79** [45] and the highly branched pentasaccharide **80** [46] (Scheme 11). The authors rationalized that the protein-detergent complexes formed were smaller with these novel arborescent neoglyco- amphiphiles. Several members of these families of detergents are commercially available [47].

## 8. Arborescent Neoglycolipids Built from Cyclic Scaffolds

Other valuable approaches to self-assembling amphiphilic and arborescent glycoarchitectures based on cyclic scaffolds such as calix[4]resorcarenes and cyclodextrins have provided interesting nanomaterials. For instance, a macrocyclic glycoconjugate having four hydrophobic undecyl chains and eight oligosaccharide moieties (oligomaltose) on the opposite sides of a calix[4]resorcarene macrocycle (**83**) has been described by the Aoyama’s group [48]. It has been prepared from a range of maltooligosaccharide lactones (**81**) and an octaamino derivative of the calix[4]resorcarene (**82**) (Scheme 13). The authors showed that they form small micelle-like nanoparticles (d = 3 nm) in water based on dynamic light scattering (DLS), gel permeation chromatography (GPC), and transmission electron microscopy (TEM). Curiously, the micellar nanoparticles agglutinated in the presence of Na_2_HPO_4_/NaH_2_PO_4_ forming aggregates up to 60–100 nm, as revealed by DLS as well as microscopy (TEM and AFM). The phosphate-induced agglutination processes could be followed by surface plasmon resonance (SPR). Kinetic analyses demonstrated that the phosphate-mediated inter(saccharide) interactions were significantly dependent on the oligosaccharide chain lengths (n), becoming more favorable with increasing n’s.

The attachment of biologically relevant carbohydrate head groups by covalent bonding in several copies and at exact positions of cyclomaltooligosaccharides (cyclodextrins, CDs) has been a highly productive and flexible strategy for the syntheses of multivalent glyconanomaterials [49]. The commercial availability of CDs in three different sizes (α-, β-, and γ-CDs) combined with their hydroxyl groups of varied accessibilities and reactivity allow excellent control of their regiochemical functionalization. ‘‘Click-type’’ ligation chemistries, including copper(I)-catalyzed azide–alkyne cycloaddition (CuAAC), thiol–ene coupling or thiourea-forming reactions, have been systematically fulfilled to secure full homogeneity of the resulting glycoconjugates. CD-based glycoconjugates constitute [49,50,51] key players in studying and understanding the fundamental structural features deciphering multivalent carbohydrate-protein recognition events [6,7,8,9,10,11,12,13,14,15]. The approach has also been applied using chemoenzymatic glycan synthesis [52]. Nanometric glycoarchitectures, endowed with the flexibility of adapting the nature and inter-saccharide distances and orientations in the presence of their cognate receptors, such as lectins or capable of mimicking the fluidity of biological membranes, have been particularly well-adapted by self-assembling amphiphilic glycans. In addition, the role played by glyconanomaterials nicely positions them toward applications in cancer therapies [53,54]. Moreover, such well-defined glycoconjugates are useful for deepening our understanding of the sugar code [55].

Amphiphilic 7-membered β-cyclodextrins with alkylthio chains at the primary-hydroxyl side and galactosylthio-oligo-(ethylene glycol) units at the most reactive secondary-hydroxyl groups, pointing the bottom segment of the cone-shaped CD, have been clear examples of the role played by a proper balancing of the hydrophilic/hydrophobic partners (Scheme 14) [56,57]. These molecules formed nanoparticles and vesicles having strong multivalent effects in their binding to the bacterial lectin PA-1L from *Pseudomonas aeruginosa*. The balance between hydrophobic and hydrophilic components in amphiphilic β-cyclodextrins, targeted by receptor specific glucoside or galactoside groups possessing either hexyl (**84**) or hexadecyl (**85**) alkyl chains have been shown to dramatically influence the structural properties of these systems. The dissimilar amphiphilic features of single cyclodextrins generated micellar aggregates and vesicles with an internal aqueous compartment able to encapsulate guests. Small-angle light scattering (SAXS), cryo-TEM and AFM investigations describe the size and shape of these self-assembling structures. Selective binding interactions with the carbohydrate moieties of the nanoassemblies by a PA-1L lectin with **85** has been successfully demonstrated using time-resolved fluorescence.

An exhaustive review on amphiphilic β-cyclodextrins harboring arborescent architectures have been described by the group of García Fernández et al. [58]. Moreover, analogous constructs were recently described using doxorubicin-loaded (Dox) glycodendrimersomes armed with mannopyranoside dendrons that are used as targeting components. They have been efficiently prepared from per-6-azido-β-cyclodextrin [59]. The amphiphilic neoglycolipids were synthesized using short propionic or valeric anhydrides at the secondary hydroxyl groups, while propargyl α-D-mannopyranoside was once again appended by azide–alkyne cycloaddition (CuAAC). Once loaded with Dox, the resulting Dox-glyconanomaterials were efficiently taken up via receptor-mediated endocytosis by MDA-MB-231 breast cancer cells that overexpress the mannose receptor. After cellular uptake, the low intracellular pH caused the release of DOX, which triggered apoptosis. Based on dynamic light scattering (DLS) measurements, the propionic anhydride-modified self-assembled formats formed nanoparticles with an average hydrodynamic size of 112 nm (PDI, 0.109) that increased to 199 nm (PDI, 0.141) after Dox-inclusion. The particle size and morphology of the liposomes were evaluated using transmission electron microscopy (TEM), which showed nanoparticles having spherical shapes with an average diameter of 45 nm. The same materials incorporating both doxorubicin and amphotericin B were used to target the delivery of the combined drugs into macrophage cells. The combination of both drugs lead to enhanced anti-leishmanial therapeutic efficacy through a synergistic effect [60].

## 9. Conclusions

The syntheses of arborescent neoglycolipids called glycodendrimersomes have been described using primarily “click-type” chemical ligations, such as the copper(I)-catalyzed azide-alkyne cycloaddition (CuAAC), radical-induced thiol-ene, aminooxylation, or thiourea ligation. They can be readily built from simple scaffolds, such as glycerol, pentaerythritol, tris(hydroxymethyl)aminomethane, clyclodextrins, and calix[n]arenes. Interesting, the chemical linkages between the hydrophilic sugar-head groups and the lipid tail seem to play a crucial role in the overall nanometer size and stability of the resulting liposomal formulations. Curiously, the data accumulated, thus far, point to an increased number of hydrophobic linkers as preferred embodiments. In addition, as seen with glycodendrimers, too many branches (generation) do not necessarily resulted in better binding carbohydrate-protein interactions. This is likely due to steric hindrances between too closely compacted sugar residues, thus, rendering lectin recognition and binding less favorable. In addition, the intrinsic mobility of the sugar head-groups can allow lipid-raft formation, thus ensuring the essential multivalent contacts with their cognate protein receptors. The design of twin-mixed glycoarchitectures is particularly appealing in this regard. Moreover, the novel dendrimersomes appeared to be more stable than their conventional counterparts and do not necessitate the use of “additional stabilizing ingredients”. Clearly, the chemical/enzymatic syntheses of arborescent neoglycolipids are much faster and simpler when compared to that of glycodendrimers which should be favored in the design of multivalent glyconanomaterials.

## Data Availability

Not applicable.

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
