# Peer review of "Recent Development in the Design of Neoglycoliposomes Bearing Arborescent Architectures"

_molecules, 2021, doi:10.3390/molecules26144281_

Round 1

Reviewer 1 Report

This is an authoritative review on dendrimer-based glycolipids that focuses on the synthesis and self-assembly properties. It is written in a very clear manner and provides an excellent information both for experts and new comers in the field. It can be accepted in essentially its present form. As a minor point, the authors might consider discussing the work of Seeberger and coworkers in (Chem. Eur. J. 2016, 22, 15216–15221 and Coll. Surf. A 2020, 598, 124804) on the applications of glycoamphiphilic cyclodextrin-based nanoparticles in drug delivery and the related review recently published in Nanomaterials (Nanomaterials 2020, 10, 2517; doi:10.3390/nano10122517).

Author Response

We are thankful to this reviewer for his very positive evaluation and for his insightful suggestions on additional uses of beta CD-glycodendrimersomes. We indeed added the 3 references suggested as [58-60] and discussed their contents accordingly marked in yellow just before the conclusion. 

Reviewer 2 Report

The review is interesting. However, a revision is needed to highlight more connection between neoglycolipids and liposomes. The description of the advances in the field of neoglycolipids is exhaustive but sometimes the relationship and the implications for the design of innovative liposomes is missing.

Moreover, glycolipids are also presented as potential detergents for membrane studies (paragraph 7) or employed for the preparation of other self-assembled scaffolds (paragraph 8), this is confusing yet in relation to liposomes.

The authors should contestualize in the introduction os in a separated paragraph the neoglycolipids in the context of liposomes design, by highlighting the advantages and their potential use and applications.

The authors should better introduce in paragaph 5 the concept of arborescent architectures and highlight the differences with "conventional glycolipids".

The authors should report recent works in which neoglycolipids bearing arborescent architectures were employed for the design of liposomes with a specific functionality.

Definitely, the authors should more focus the review on liposome design using neoglycolipids, as indicated in the title, by addressing the comments above, or, alternatively, change the title and re-organize some parts of the review and not stressing in a relevant manner on the potential use on liposome formulation.

Author Response

We have appreciated the suggestions made by this reviewer as they will enhance the quality of the review.

As per the 1st comment:

The review is interesting. However, a revision is needed to highlight more connection between neoglycolipids and liposomes. The description of the advances in the field of neoglycolipids is exhaustive but sometimes the relationship and the implications for the design of innovative liposomes is missing.

We added text from lines 74-85 to better highligth the connection in paragraph 2 marked in yellow.

For the comment :

"Moreover, glycolipids are also presented as potential detergents for membrane studies (paragraph 7) or employed for the preparation of other self-assembled scaffolds (paragraph 8), this is confusing yet in relation to liposomes."

We included these détergents simply as "additional" examples of the arborescent strategy design, obviously applicable to liposomal formulations.

We beleive that the remaining suggestions are properly addressed from the above made modifications.

For the "Relationship", we added text in lines 128-135 of paragraph 4.

Round 2

Reviewer 2 Report

Manuscript is ready for publication